# Numerical Study of the Influence of Fishnet Mesh Size on a Floating Platform

**Hung-Jie Tang** [1] **, Chai-Cheng Huang** [2] **and Ray-Yeng Yang** [3,*]

[1]  Tainan Hydraulics Laboratory, National Cheng Kung University, Tainan 709, Taiwan;
    hjtang@thl.ncku.edu.tw
[2]  Department of Marine Environment and Engineering, National Sun Yat-sen University, Kaohsiung 804,
    Taiwan; cchuang@mail.nsysu.edu.tw
[3]  Department of Hydraulic and Ocean Engineering, National Cheng Kung University, Tainan 701, Taiwan
[*]  Correspondence: ryyang@mail.ncku.edu.tw; Tel.: +886-6-2757275 (ext. 63246)

**Abstract:** This study aims to investigate the influence of fishnet mesh size on a floating platform. A self-developed, time-domain numerical model was used for the evaluation. This model is based on potential flow theory, uses the boundary element method (BEM) to solve nonlinear wave-body interactions, and applies the Morison equation to calculate the hydrodynamic forces exerted on fishnets. The mooring system is treated as a linear and symmetric spring. The results near the resonant frequency of the platform indicate that the smaller the fishnet mesh size, the lower the heave, pitch, and sea-side tension response amplitude operators (RAOs), but the higher the reflection coefficient. The results in the lower frequency region reveal that the smaller the fishnet mesh size, the lower the surge and heave RAOs, but the higher the pitch and tension RAOs. Meanwhile, the time-domain results at the resonant frequency of heave motion are shown to indicate the influences of a platform with various fishnets mesh sizes on the rigid body motion, mooring line tension, and transmitted wave heights. In addition, a comparison of nonlinear effects indicates that, after reducing the fishnet mesh size, the second-order RAOs of heave, pitch, and sea-side tension decrease, but the changes are minor against the first-order results.

**Keywords:** floating platform; fishnet mesh size; frequency-domain; time-domain; nonlinear waves; BEM

## 1. Introduction

In recent years, owing to environmental impacts and spatial conflicts with other industries, the development of marine cage aquaculture moved towards deep sea operations. For the better utilization of ocean space, as well as reduced construction costs, the multi-purpose floating platform is becoming a popular research topic around the world. For example, a European Union (EU) project called "The Blue Growth Farm" was proposed to develop a multi-purpose floating platform that is intended to combine blue energy (renewable energy) and fish farming and use renewable energy to supply the power required for intelligent farming equipment. Nevertheless, the fishnet used has not yet been addressed in current research ([1,2]).

In recent decades, many researchers adopted the Morison-type numerical model to study the hydrodynamic characteristics of marine fish cages [3–10]. This net cage structure has been treated as a so-called small-body in order to ignore the wave–body interaction. Conversely, many studies considered the floating structure as a so-called large-body in order to analyze the nonlinear wave–body interaction by means of potential flow theory [11–15]. However, studies of marine aquaculture structure that include small-body and large-body structures remain few and far [16,17]. Therefore, a number

of issues that must be resolved still exist. For example, the influences of different fishnet mesh sizes on marine cage systems have been investigated [18,19]. Moreover, fishnet biofouling in the real sea is inevitable and can result in the fishnet exhibiting a narrowed mesh size and significant mass increment [20,21]. In the real world, replacing the fishnet with different mesh sizes according to the size of fish during fish farming is also necessary. Therefore, studying the effect of fishnet mesh size on floating platform dynamics is important.

In our previous study [16], a two-dimensional, nonlinear numerical wave tank incorporating potential flow theory was developed to investigate the dynamic interaction between waves and a floating platform with a fishnet. This model has been validated by physical model tests and shows good agreement. The research [16] studied related conditions such as net depth, net width, and the nonlinearity of dynamic response. In this study, we continue to explore the impact of fishnet mesh size on the hydrodynamic characteristics of an aquaculture platform, with a view of providing a reference for the development of a net-type aquaculture floating platform.

## 2. Description of the Numerical Model

An aquaculture-purposed floating platform (see Figure 1a) consisting of a pair of floating rectangular pontoons and restrained by a linear symmetric mooring system is shown in Figure 1b, where $a$ is the width of each pontoon, $b$ is the spacing between the two pontoons, $d$ is the draught, $(x_G, z_G)$ is the position of the center of gravity, $l_G$ is the pitch moment arm, $\theta_0$ is the mooring line angle, and $l_0$ is the original length of the mooring line. The floating structure was deployed in a numerical wave tank with a constant water depth, $h$. A numerical damping zone was used at each end of the wave tank to dissipate the reflected and transmitted waves, where $x_{d1}$ and $x_{d2}$ are the entrance positions.

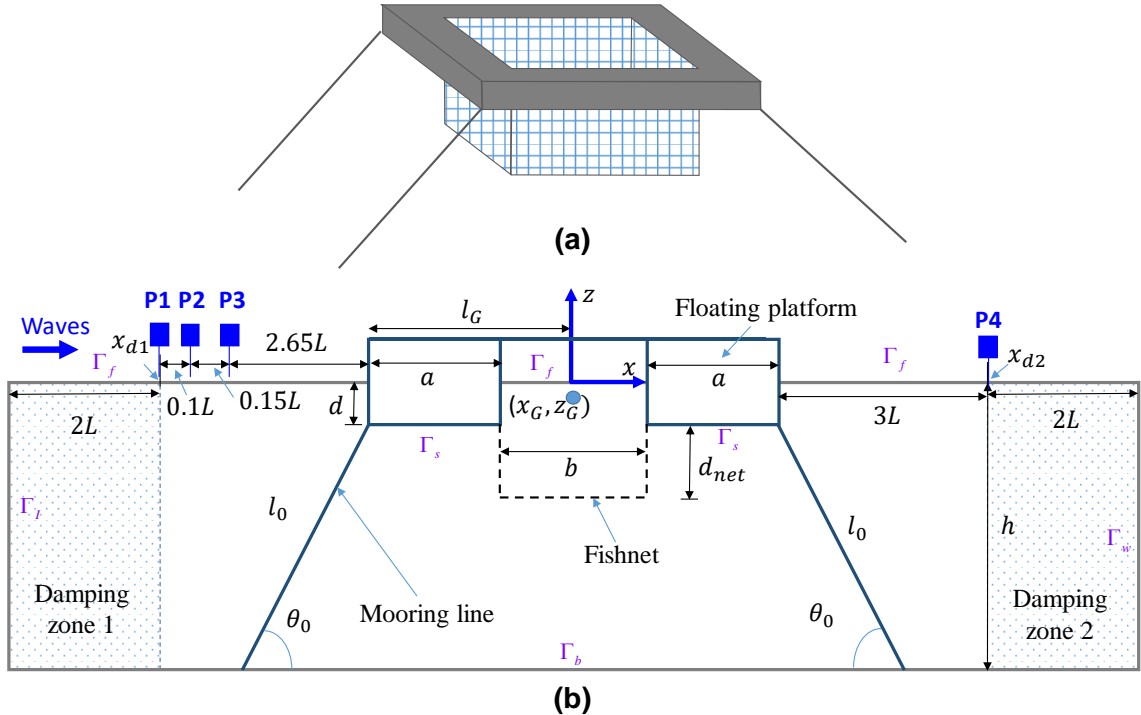

**Figure 1.** (**a**) The concept design and (**b**) definition sketch of the floating platform with a fishnet in a numerical wave tank.

## 2.1. Governing Equation

The two-dimensional flow field is assumed to be incompressible, inviscid, and irrotational. Thus, a velocity potential exists and satisfies the Laplace equation:

$$\nabla^2 \phi = \frac{\partial^2 \phi}{\partial x^2} + \frac{\partial^2 \phi}{\partial z^2} = 0. \tag{1}$$

Incorporating Equation (1) into the Green second identity, the velocity potential in the fluid domain can be determined by solving the following boundary integral equation (BIE):

$$\alpha \phi_i = \int_{\Gamma_j} \left( \frac{\partial G_{ij}}{\partial n} \phi_j - G_{ij} \frac{\partial \phi_j}{\partial n} \right) d\Gamma_j \tag{2}$$

where $G_{ij} = \ln r_{ij}/2\pi$ is the fundamental solution to the Laplace equation and represents a flow field generated by a concentrated unit source acting at the $i$th source point, $r_{ij}$ is the distance from source point $(x_i, z_i)$ to field point $(x_j, z_j)$, and $\alpha$ is the internal solid angle between the two boundary elements. In this model, the linear element scheme and six-point Gaussian quadrature integration method are applied to solve the BIE.

## 2.2. Inflow Boundary Condition

On the basis of the continuity of velocity, a theoretical particle velocity profile can be used to specify the boundary value along the inflow boundary. For nonlinear regular waves, the second-order Stokes wave is used to prevent a mismatch between the input velocity profiles and real water particle velocity, as described in [22,23] and expressed below:

$$\frac{\partial \phi}{\partial n} = - \left[ \begin{array}{c} \frac{gAk}{\sigma} \frac{\cosh k(z+h)}{\cosh kh} \cos(kx - \sigma t) \\ + \frac{3}{4} A^2 k\sigma \frac{\cosh 2k(z+h)}{\sinh^4 kh} \cos 2(kx - \sigma t) \end{array} \right] f_m \quad \text{on } \Gamma_I \ , \tag{3}$$

where $A$, $k$, and $\sigma$ are the amplitude, wave number, and angular frequency, respectively; $g$ is the gravitational acceleration; and $t$ is the time. The modulation function $f_m$ is used to prevent impulse-like behavior of a wave maker and is written as

$$f_m(t) = \begin{cases} \frac{1}{2} \left[ 1 - \cos(\frac{\pi t}{T_m}) \right] & \text{for } t < T_m, \\ 1 & \text{for } t \geq T_m, \end{cases} \tag{4}$$

where $T_m$ is the modulation duration that depends on wave steepness. For a steeper wave, the modulation duration is usually twice as long as a regular wave period.

## 2.3. Free Surface Boundary Condition

One of the most popular and successful approaches to the fully nonlinear free surface simulation is the mixed Eulerian and Lagrangian (MEL) method, which was first presented by Longuet–Higgins and Cokelet [24]. In this method, the kinematic and dynamic free surface boundary conditions are transformed into the Lagrangian framework. To obtain numerical solutions for wave propagation in a wave tank, the scheme used numerical damping zones at both ends of the wave tank to absorb the transmitted wave energy at the end of the tank, as well as to dissipate the reflected waves in front of the input boundary. The numerical damping zones [25,26] are incorporated into the free surface boundary conditions as

$$\begin{cases} \frac{dx}{dt} = \frac{\partial \phi}{\partial x} \\ \frac{dz}{dt} = \frac{\partial \phi}{\partial z} - \nu(x)(z - z_e) \\ \frac{d\phi}{dt} = -gz + \frac{1}{2}|\nabla\phi|^2 - \nu(x)(\phi - \phi_e) \end{cases} \quad \text{on } \Gamma_f, \tag{5}$$

where $v(x)$ is the damping coefficient of the numerical damping zone, given by

$$v(x) = \begin{cases} \alpha_d \sigma[(x_{d1} - x)/L]^2 & x \leq x_{d1}, \\ \alpha_d \sigma[(x - x_{d2})/L]^2 & x \geq x_{d2}, \\ 0 \end{cases} \tag{6}$$

where $\alpha_d$ is the dimensionless parameter for the strength of the damping zone; after several tests, we found that $\alpha_d$ set to 1 is adequate for obtaining accurate results. $L$ is the wavelength of the input wave, while $x_{d1}$ and $x_{d2}$ are the entrance positions of each damping zone shown in Figure 1b. Meanwhile, $z_e$ and $\phi_e$ in Equation (5) are the entrance wave elevation and potential function in the front damping zone, only existing in $x \leq x_{d1}$. Tanizawa [26] applied this damping zone technique to dissipate the wave energy reflected from the structure, but without disturbing the outgoing incident waves. For practical purposes, the nonlinear analytical solution of the second-order Stokes wave theory was adopted in damping zone 1 to improve the computational process. In this model, the entrance potential and wave elevation are written as follows:

$$\begin{cases} \phi_e = \frac{Ag}{\sigma} \frac{\cosh k(z+h)}{\cosh kh} \sin(kx - \sigma t) + \frac{3}{8} A^2 \sigma \frac{\cosh 2k(z+h)}{\sinh^4 kh} \sin 2(kx - \sigma t), \\ z_e = A\cos(kx - \sigma t) + \frac{kA^2 \cosh kh}{4\sinh^3 kh}(2 + \cosh 2kh)\cos 2(kx - \sigma t). \end{cases} \tag{7}$$

Additionally, the nodal velocities in Equation (5) are obtained by using the cubic spline scheme in the curvilinear coordinate system, as described in Section 2.6. The corner problem between the free surface and body surface is treated according to [27], as described in Section 2.7.

## 2.4. Body Surface Boundary Condition

In this model, the body surface ($\Gamma_s$) is impermeable. Therefore, the fluid velocity is equal to the normal velocity on the body surface:

$$\frac{\partial \phi}{\partial n} = n_1 \dot{x}_G + n_2 \dot{z}_G + n_3 \dot{\theta}_G \quad \text{on } \Gamma_s, \tag{8}$$

where $(n_1, n_2, n_3) = (n_x, n_z, r_z n_x - r_x n_z)$ is the unit normal vector on the body boundary, and $(r_x, r_z) = (x - x_G, z - z_G)$ is the position vector from the body surface to the gravity center. Subscript $G$ designates the gravity center of the body; $(\dot{x}_G, \dot{z}_G)$ are the translational velocities in the $x$ and $z$ axes (surge and heave motions), respectively; and $\dot{\theta}_G$ is the angular velocity about the $y$ axis (pitch motion).

## 2.5. Rigid Boundary Condition

At the end-wall ($\Gamma_w$) and bottom ($\Gamma_b$) of the wave tank, the boundary conditions are considered impermeable. The normal velocities are then set to zero

$$\frac{\partial \phi}{\partial n} = 0 \text{ on } \Gamma_b \text{ and } \Gamma_w. \tag{9}$$

## 2.6. Curvilinear Coordinate System

In this paper, a curvilinear coordinate system and the cubic spline scheme are adopted to solve the spatial derivatives of velocity potential in Equation (5) on the free surface boundary. The relationship between the velocity components in Cartesian and curvilinear coordinates is written as

$$\begin{cases} \frac{\partial \phi_f}{\partial x} = \frac{\partial \phi_f}{\partial s}\cos \beta_f - \frac{\partial \phi_f}{\partial n}\sin \beta_f, \\ \frac{\partial \phi_f}{\partial z} = \frac{\partial \phi_f}{\partial s}\sin \beta_f + \frac{\partial \phi_f}{\partial n}\cos \beta_f, \end{cases} \tag{10}$$

where $\phi_f$ represents the potential function on the free surface and $\beta_f$ is the angle between $s$, a section of the free surface, and the $x$ axis. The normal velocities of the free surface $\partial\phi_f/\partial n$ are obtained after solving the BIE, while the angle $\beta_f$ is determined from the following equation:

$$\tan\beta_f = \frac{\sin\beta_f}{\cos\beta_f} = \frac{\partial z/\partial s}{\partial x/\partial s}, \tag{11}$$

where, $\partial\phi_f/\partial s$, $\partial x/\partial s$, and $\partial z/\partial s$ are calculated by using cubic spline interpolation in curvilinear coordinates along the free surface.

Once the values of the time derivative of the potential function on the right side of Equation (5) are known, the substantial derivative equations on the left side can be used to predict the new nodal position and its corresponding potential on the free surface boundary by employing the fourth-order Runge–Kutta (RK4) method as a time marching scheme. This process was repeated until the simulation reached a steady-state condition.

Note that the node-regridding and smoothing technique is also applied in the present model using the cubic spline interpolation on the curvilinear coordinate system in order to prevent free surface nodes from moving too close to one another and to prevent the occurrence of the saw-tooth condition, which may lead to numerical instability.

### 2.7. Corner Problem between the Free Surface and Body Surface

At the intersection of the body surface and free surface, the discontinuity of the flux occurs as a result of the discontinuity of the normal direction. Although the cubic spline scheme is accurate for determining the tangential slope at the end-point with the natural condition (curvature equal to zero) and the Lagrangian polynomial method, the requirement of continuity of flux at the corner is still difficult to achieve. To deal with this discontinuity, the double collocation node technique is often used. Grilli and Svendsen [27] proposed a treatment for the corner problem at the intersection based on the continuity flux as follows:

$$\frac{\partial\phi_f}{\partial s} = \frac{\partial\phi_f}{\partial n}\frac{\cos(\beta_b - \beta_f)}{\sin(\beta_b - \beta_f)} - \frac{\partial\phi_b}{\partial n}\frac{1}{\sin(\beta_b - \beta_f)}, \tag{12}$$

where the subscripts $b$ and $f$ denote the body and water free surface, respectively. $\partial\phi_b/\partial n$ and $\partial\phi_f/\partial n$ are the normal velocities on the free surface and body surface, respectively; and $\partial\phi_f/\partial s$ is the modified tangential velocity on the free surface and will be used in Equation (10) when dealing with the corner problem. In this model, the input boundary angle at the front of the tank is $\beta_b = \pi/2$, while that of the wall boundary at the end of the wave tank is $\beta_b = 3\pi/2$.

### 2.8. Wave Forces on the Body

The hydrodynamic forces on the body can be calculated by integrating the pressure around the wetted body surface as

$$\begin{cases} \boldsymbol{F} = \int_{\Gamma_s} -\rho\left(\phi_t + gz + \frac{1}{2}|\nabla\phi|^2\right)\boldsymbol{n}\mathrm{d}s, \\ \boldsymbol{M} = \int_{\Gamma_s} -\rho\left(\phi_t + gz + \frac{1}{2}|\nabla\phi|^2\right)\boldsymbol{r}\times\boldsymbol{n}\mathrm{d}s, \end{cases} \tag{13}$$

where $\boldsymbol{F}$ and $\boldsymbol{M}$ are the hydrodynamic force and moment, respectively; $\rho$ is the water density; $\boldsymbol{n}$ is the normal unit vector on the body surface and points into the body; and $\boldsymbol{r}$ is the position vector from the gravity center to body's surface.

### 2.9. Acceleration Potential Method

In order to evaluate the hydrodynamic forces on the floating body using Equation (13), both gradients of velocity potential ($\nabla\phi$) and unsteady $\phi_t$ terms on the wetted body surface must be

determined beforehand. $\nabla\phi$ is evaluated by the regular boundary element method (BEM), while $\phi_t$ is determined by an acceleration potential method proposed by [28], taking the advantage of the feature that $\phi_t$ also satisfies the Laplace equation:

$$\nabla^2\phi_t = \frac{\partial^2\phi_t}{\partial x^2} + \frac{\partial^2\phi_t}{\partial z^2} = 0. \tag{14}$$

In accordance with [27], the body surface–surface boundary condition in the acceleration field is described as

$$\frac{\partial\phi_t}{\partial n} = n_1\ddot{x}_G + n_2\ddot{z}_G + n_3\ddot{\theta}_G + q \quad \text{on } \Gamma_s, \tag{15}$$

where $(\ddot{x}_G, \ddot{z}_G)$ are the translational accelerations in the $x$- and $z$-axes, and $\ddot{\theta}_G$ is the angular acceleration around the $y-$axis. In turn, $q$ is defined as

$$\begin{aligned}
q = {} & n_1\dot{\theta}_G\left(r_x\dot{\theta}_G - 2\dot{z}_G + 2\frac{\partial\phi}{\partial z}\right) + n_2\dot{\theta}_G\left(r_z\dot{\theta}_G + 2\dot{x}_G - 2\frac{\partial\phi}{\partial x}\right) \\
& + k_n\left[\left(\frac{\partial\phi}{\partial x} - \dot{x}_G - \dot{\theta}_G r_z\right)^2 + \left(\frac{\partial\phi}{\partial z} - \dot{z}_G + \dot{\theta}_G r_x\right)^2\right] \\
& - k_n\left[\left(\frac{\partial\phi}{\partial x}\right)^2 + \left(\frac{\partial\phi}{\partial z}\right)^2\right] - \frac{\partial\phi}{\partial s}\frac{\partial^2\phi}{\partial n\partial s} + \frac{\partial\phi}{\partial n}\frac{\partial^2\phi}{\partial s^2},
\end{aligned} \tag{16}$$

where $k_n = 1/\rho^*$ is the normal curvature along the $s$ direction of the body surface and $\rho^*$ is the radius of the curvature. Note that $\partial\phi/\partial s$, $\partial^2\phi/(\partial s\partial n)$, and $\partial^2\phi/\partial s^2$ are calculated using cubic spline interpolation in the curvilinear coordinates along the body surface. The above variables at the corners of the structure are considered the natural condition (curvature equal to zero), while those between the structure and free surface are modified by the same method, as described in Section 2.7.

For solving $\phi_t$ in the acceleration field, four methods are available, which include (1) the iterative method [29,30], (2) the modal decomposition method [31], (3) the implicit boundary condition method [28], and (4) the indirect method [32]. Detailed descriptions of these can be found in [22,33,34].

*2.10. Modal Decomposition Method*

The modal decomposition method was first introduced in [31]. This approach solves the BIE for the acceleration field. The acceleration potential function is decomposed into four modes corresponding to three unit accelerations for surge-heave-pitch motions (radiation problem) and acceleration due to the incident wave field (diffraction problem). Using these four modes in Equation (17) and the boundary conditions listed in Equations (18)–(21), the unknown mode's amplitude can be determined by solving its respective BIE. The $\phi_t$ is given by

$$\phi_t = \sum_{m=1}^{3} a_m\varphi_m + \varphi_4, \tag{17}$$

where $a_m$ is the $m$th mode component of generalized body acceleration (1 = surge, 2 = heave, 3 = pitch, 4 = diffraction mode).

The boundary conditions in the acceleration field for each mode are given as

$$\frac{\partial\varphi_m}{\partial n} = \begin{cases} n_m & m = 1,2,3 \\ q & m = 4 \end{cases} \quad \text{on } \Gamma_s, \tag{18}$$

$$\varphi_m = \begin{cases} 0 & m = 1,2,3 \\ -gz - \frac{1}{2}|\nabla\phi|^2 & m = 4 \end{cases} \quad \text{on } \Gamma_f, \tag{19}$$

$$\frac{\partial \varphi_m}{\partial n} = 0 \ (m = 1, 2, 3) \quad \text{on } \Gamma_I, \tag{20}$$

$$\frac{\partial \varphi_m}{\partial n} = 0 (m = 1 \sim 4) \quad \text{on } \Gamma_b \text{ and } \Gamma_w. \tag{21}$$

The inflow boundary condition for mode 4 is obtained from the second-order Stokes wave theory:

$$\frac{\partial \varphi_4}{\partial n} = - \left[ \begin{array}{l} gAk \frac{\cosh k(z+h)}{\cosh kh} \sin(kx - \sigma t) \\ + \frac{3}{2} A^2 k \sigma^2 \frac{\cosh 2k(z+h)}{\sinh^4 kh} \sin 2(kx - \sigma t) \end{array} \right] \quad \text{on } \Gamma_I. \tag{22}$$

After solving the BIE for each mode, the values of $\varphi_m$ and $\partial \varphi_m / \partial n$ for all of the boundaries are obtained, with the remaining unknown in Equation (17) being $a_m$.

Substituting Equation (17) into Equation (13), combined with Newton's second law including the wave hydrodynamic and other forces (with subscript $e$; for example, gravitational force, damping force, restoring force, and force on net), the equation of motion becomes

$$\begin{cases} ma_1 = \int_{\Gamma_s} -\rho \Big( a_1 \varphi_1 + a_2 \varphi_2 + a_3 \varphi_3 + \varphi_4 + gz + \frac{1}{2} |\nabla \phi|^2 \Big) n_1 ds + F_{ex}, \\ ma_2 = \int_{\Gamma_s} -\rho \Big( a_1 \varphi_1 + a_2 \varphi_2 + a_3 \varphi_3 + \varphi_4 + gz + \frac{1}{2} |\nabla \phi|^2 \Big) n_2 ds + F_{ez}, \\ I_G a_3 = \int_{\Gamma_s} -\rho \Big( a_1 \varphi_1 + a_2 \varphi_2 + a_3 \varphi_3 + \varphi_4 + gz + \frac{1}{2} |\nabla \phi|^2 \Big) n_3 ds + M_{ey}. \end{cases} \tag{23}$$

After solving Equation (23), the generalized acceleration $a_m$ can be obtained. The details of the extra forces are described below.

### 2.11. Mooring Force

The mooring system is considered linear and symmetrical (see Figure 1b) with the spring constant $K$, while the wave hydrodynamic forces on the mooring line are negligible in comparison with the force exerted on the floating dual pontoon. The pre-tension force of this mooring system is written as

$$F_{T_0} = \frac{2\rho g a d\lambda - mg}{2 \sin \theta_0}, \tag{24}$$

where $\lambda$ is the total length of the floating dual pontoon structure in the direction of $y$.

### 2.12. Wave Forces on Fishnet

In this model, a fishnet is set up between the pontoons and secured by a steel frame. The net is assumed not to deform. By applying the lumped mass method discussed in [5], the fishnet panel is divided into several elements and nodes, while a modified Morison equation [35] to calculate the drag and inertia forces on the net elements is given as

$$F_{\text{net}} = \frac{1}{2} \rho C_D A_{\text{net}} V_R |V_R| + \rho \delta_{\text{net}} C_M \frac{dV}{dt} - \rho \delta_{\text{net}} K_M \frac{d\dot{R}}{dt}, \tag{25}$$

where $\rho$ is water density, $C_D$ is the drag coefficient, $C_M = 1 + K_M$ is the inertia coefficient, and $K_M$ is the added mass coefficient. In this model, $C_M = 2.0$ is assumed, which is generally between 1.0 and 2.0. $A_{\text{net}}$ is the projected area of the net element, $\delta_{\text{net}}$ is the volume of the net element, and $V_R = V - \dot{R}$ the relative velocity between the flow field and net element. In turn, $V$ is the flow velocity at the center of a net element, $\dot{R}$ is the central velocity of the net element, $dV/dt$ is the fluid particle acceleration at the center of the net element, and $d\dot{R}/dt$ is the central acceleration of the net element. Details about fluid particle velocity and acceleration are described in [16].

According to Loland's empirical formula [36], the drag force, which is parallel to the fluid motion, and the lift force, which is perpendicular to the fluid motion, are as follows:

$$\begin{cases} F_D = \frac{1}{2}\rho C_D(\alpha)A_{\text{net}}|V_R|^2, \\ F_L = \frac{1}{2}\rho C_L(\alpha)A_{\text{net}}|V_R|^2, \end{cases} \tag{26}$$

where $C_D(\alpha)$ and $C_L(\alpha)$ are coefficients related to the angle of $\alpha$ between the fluid particle velocity vector and the normal vector of the net element:

$$\begin{cases} C_D(\alpha) = 0.04 + (-0.04 + 0.33S_n + 6.54S_n{}^2 - 4.88S_n{}^3)\cos(\alpha), \\ C_L(\alpha) = (-0.05S_n + 2.3S_n{}^2 - 1.76S_n{}^3)\sin(2\alpha), \end{cases} \tag{27}$$

where $S_n$ is the solidity ratio, which is defined as the ratio between the area covered by the threads and the total area of the net panel.

### 2.13. Consideration of Damping Effects

Experimental testing in a physical wave tank revealed that dynamic responses near the resonant frequency of body motions significantly dampen; similar phenomena were also identified by [37] and may be attributed to the fluid viscous effect. The flow field is assumed to be inviscid in the present model and, at present, determining the damping coefficients of a floating structure is nearly impossible. To simplify this problem, an uncoupled damping coefficient matrix is incorporated into the equation of motion of Equation (22) to represent the damping forces. The equation is then rewritten as

$$\begin{bmatrix} m & & \\ & m & \\ & & I_G \end{bmatrix}\begin{bmatrix} \ddot{x}_G \\ \ddot{z}_G \\ \ddot{\theta}_G \end{bmatrix} + \begin{bmatrix} C_{xx} & & \\ & C_{zz} & \\ & & C_{\theta\theta} \end{bmatrix}\begin{bmatrix} \dot{x}_G \\ \dot{z}_G \\ \dot{\theta}_G \end{bmatrix} + [K]\begin{bmatrix} x_G \\ z_G \\ \theta_G \end{bmatrix} = \begin{bmatrix} F_x \\ F_z \\ M_y \end{bmatrix}, \tag{28}$$

where $[K]$ is the stiffness matrix, and the components are determined numerically at each time step according to the mooring angle. $F_x$, $F_z$, and $M_y$ are the resultant hydrodynamic forces acting on the platform, respectively, while $C_{xx}$, $C_{zz}$, and $C_{\theta\theta}$ are the damping coefficients of surge, heave, and pitch motion, respectively. Furthermore, these damping coefficients are assumed to be equal and denoted by $C$, and are obtained from the damping ratio [38]:

$$\zeta = \frac{C}{2\sqrt{Km}}. \tag{29}$$

In this study, the damping ratio is set to 0.1 in order to fit our experimental data, while $K$ is the spring constant and $m$ the total mass of the floating structure.

## 3. Numerical Model Test

In our previous research [16], the numerical model under discussion here was verified by physical model tests in a wave flume. Table 1 lists the physical model data of the aquaculture platform and its corresponding materials. According to previous results [16], apart from the resonant frequency region, the influence of the fishnet on the platform is small. The conclusion was that the mass and drag force of the fishnet is too small to affect the motion of the platform. Thus, in this study, we decrease the fishent mesh size to increase the total mass and drag force, but keep the twine diameter constant. Table 2 shows the mesh size, the solidity ratio [36], the total mass of the fishnet (obtained according to the solidity ratio), and the mass ratio between the fishnet and platform.

**Table 1.** Characteristics of the floating platform with a fishnet.

| Specifications | Sizes |
|---|---|
| Wave amplitude ($A$) | 0.02 m |
| Wave period ($T$) | 0.73–2.54 s |
| Water depth ($h$) | 0.80 m |
| Total mass of platform ($m$) | 62.58 kg |
| Width of platform ($a$) | 0.25 m |
| Spacing between pontoons ($b$) | 0.50 m |
| Draft ($d$) | 0.153 m |
| Pitch moment arm ($l_G$) | 0.50 m |
| Moment of inertia ($I_G$) | 8.93 kg·m$^2$ |
| Spring constant (K) | 674.93 N/m |
| Mooring angle ($\theta_0$) | 52° |
| Gravity center ($x_G, z_G$) | (0.0 m, −0.0861 m) |
| Net depth ($d_{net}$) | 0.455 m |
| Twine diameter ($D_{line}$) | 0.175 cm |
| Specific gravity of net (Nylon) | 1.14 |

**Table 2.** Numerical model test conditions.

| Half Mesh Size (cm) $\lambda$ | Solidity Ratio $S_n$ | Total Mass (kg) $m_{net}$ | Mass Ratio (%) $m_{net}/m$ |
|---|---|---|---|
| 2.0 | 0.179 | 1.645 | 2.6 |
| 1.0 | 0.365 | 3.361 | 5.4 |
| 0.5 | 0.761 | 7.003 | 11.2 |

In addition, the appropriate grid size and time step for the numerical model are obtained through a preliminary convergence test. The grid size is $L/30$ ($L$ is the wave length) on the free surface, the space between each pontoon is 10 elements, $h/20$ on both sides of the NWT, $L/10$ on the bottom, and 40 elements are present on each pontoon surface. The appropriate marching time step of the RK4 is $T/32$, while the total simulated time is $50T$. The input wave periods are in the range of 0.73 s and 2.54 s, while the input wave heights are all 4 cm. The simulation was executed on a personal computer (Intel i5 CPU, 16 GB RAM), with a calculation time for each simulation of about 30 min.

## 4. Results and Discussion

### 4.1. Frequency-Domain Results

In general, the response amplitude operator (RAO) is used to describe the first-order dynamic response of body motions related to incident wave amplitudes through an FFT (Fast Fourier Transform) analysis, as is shown in Table 3. The second-order dynamic responses can also be calculated using the same analysis, as the dynamic responses of wave-body interactions are harmonic. Therefore, in this study, the second-order RAOs are adopted to describe the nonlinear dynamic responses. The incident and reflected waves are separated by Mansard and Funke's method [39] using three wave gauges installed in front of the structure (see Figure 1b).

**Table 3.** Definition of the response amplitude operators (RAOs) and frequency.

| Normalized Parameters | Definition |
| --- | --- |
| Normalized surge RAO | $x_G/A$ |
| Normalized heave RAO | $z_G/A$ |
| Normalized pitch RAO | $l_G\theta_G/A$ |
| Normalized tension RAO | $F_T/KA$ |
| Normalized angular frequency | $\sigma^2 h/g$ |

Figures 2–4 show a comparison of surge, heave, and pitch RAO for a platform with different fishnet mesh sizes ($\lambda$). The results include previous measurements and simulations from [16] to ensure that the present simulations fall within a reasonable area. Unsurprisingly, the results indicate that, when the fishnet mesh sizes are different, the RAOs of platform motion will change significantly. The surge RAO exhibits good reduction in the low-frequency region ($\sigma^2 h/g < 2$), as $\lambda$ decreases from 2 cm to 0.5 cm. The low-frequency response is usually related to the restoring mooring force, except that the differences in surge of the RAO among the simulated cases are small. As for the heave RAO, in both the low-frequency region ($\sigma^2 h/g < 2$) and the resonant frequency region (near $\sigma^2 h/g = 3.9$), the heave RAO results are greatly reduced with decreasing $\lambda$. Moreover, a slight phase difference of heave RAO can be observed in the resonant frequency region. Finally, for the pitch RAO, the response increases greatly near $\sigma^2 h/g = 2$, but decreases in the frequency region between $\sigma^2 h/g = 3$ and 6. Overall, reducing the mesh size of the net is helpful for mitigating the dynamic response.

Figure 5 presents a comparison of sea-side tension RAO across different fishnet mesh sizes ($\lambda$). Because the mooring system is deployed to limit the movement of the platform, the response of the tension RAO should be closely related to the surge-heave-pitch motion. Around $\sigma^2 h/g = 4$, the response of the tension RAO may be dominated by the heave RAO. In this region, the tension RAO decreases with $\lambda$, but no phase difference occurs that can be observed in the heave RAO. In the region of $\sigma^2 h/g < 2$, the response of the tension RAO relates to the combination of surge-heave-pitch motion, but the pitch RAO seems to dominate the tension response. In this region, the tension RAO generally increases as $\lambda$ decreases.

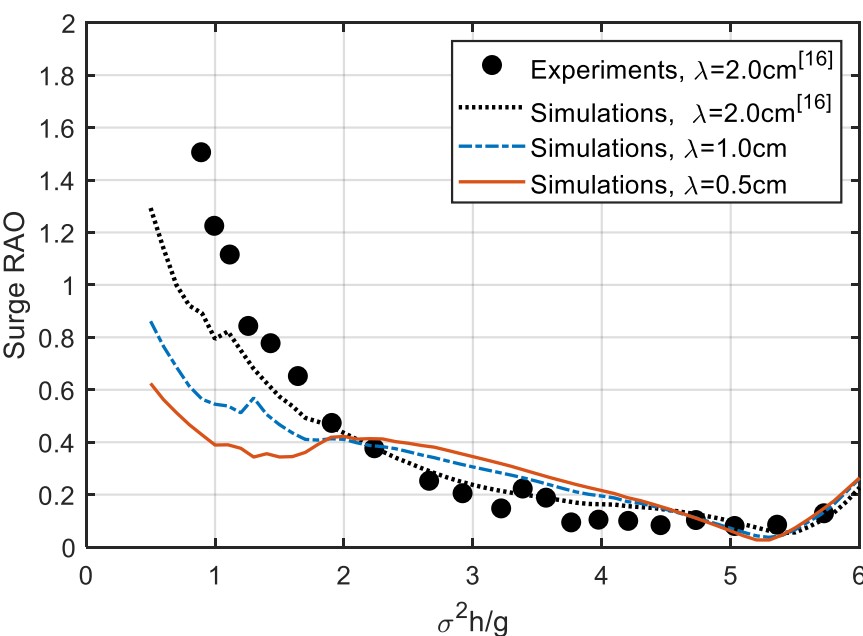

**Figure 2.** Comparison of the surge response amplitude operator (RAO) for the platform with various fishnet mesh sizes.

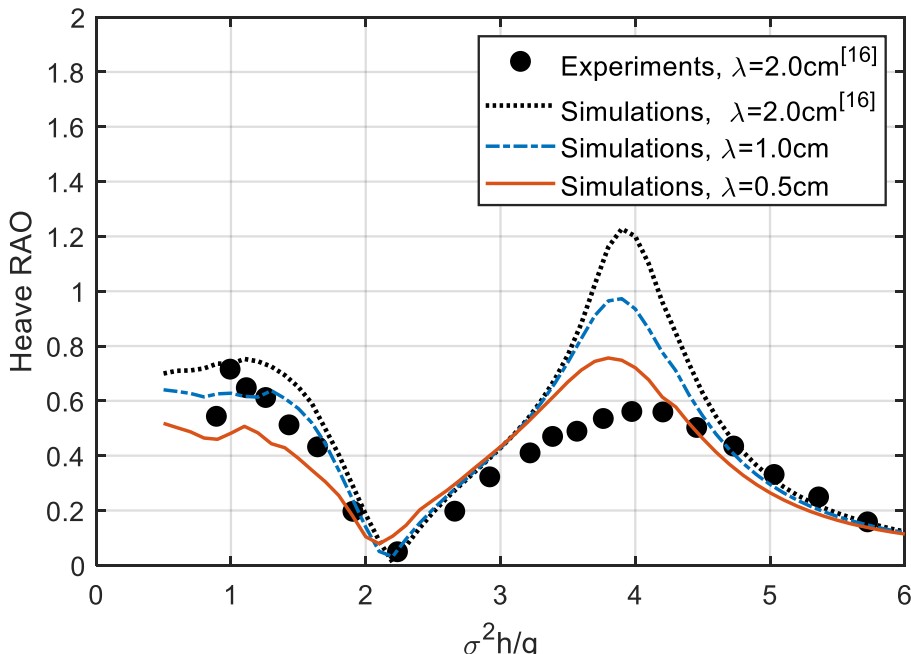

**Figure 3.** Comparison of the heave RAO for the platform with various fishnet mesh sizes.

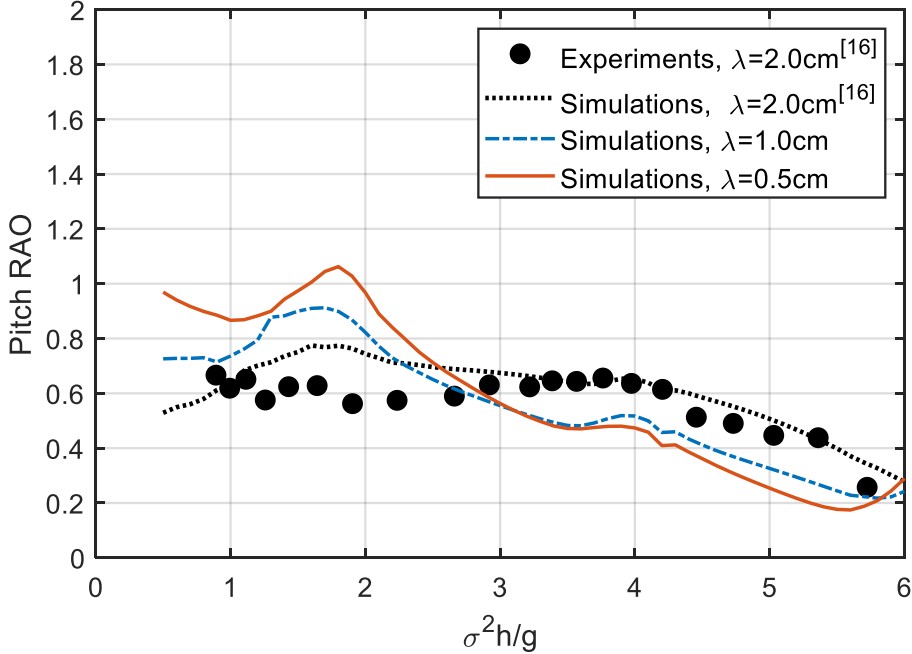

**Figure 4.** Comparison of the pitch RAO for the platform with various fishnet mesh sizes.

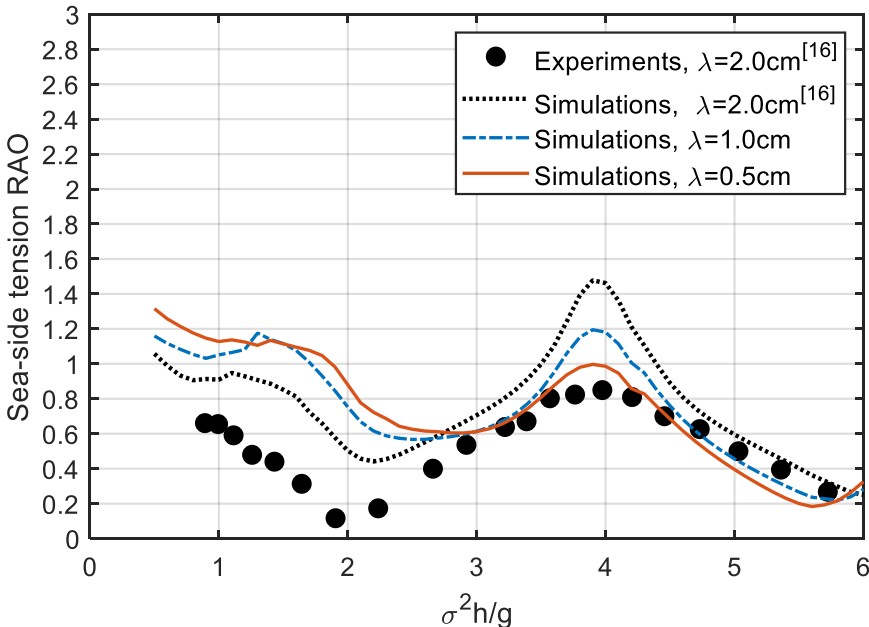

**Figure 5.** Comparison of the sea-side tension RAO for the platform with various fishnet mesh sizes.

In some studies [40,41], the floating structure is used as the floating breakwater to protect the facilities behind it. Similarly, this aquaculture-purposed platform may demonstrate good performance as a breakwater. Furthermore, the drag force on the net and its mass may affect wave–platform interactions, in spite of the fact that the wave–net interaction is ignored in this model. Figure 6 shows a comparison of the reflection coefficient under different fishnet mesh sizes ($\lambda$). In addition, the result includes previous data from [16] in order to compare with the present simulations. A remarkable increment in the reflection coefficient is found in the range of $\sigma^2 h/g$ from 3.7 to 6.0 when $\lambda$ decreases from 2 cm to 0.5 cm, especially in the resonant frequency region of heave motion ($\sigma^2 h/g = 3.9$). However, a not to be ignored reduction in the reflection coefficient appears in the range of $\sigma^2 h/g$, from 2.0 to 3.7. Overall, the platform with a fishnet with a smaller mesh size exhibits better performance in terms of the reflection coefficient.

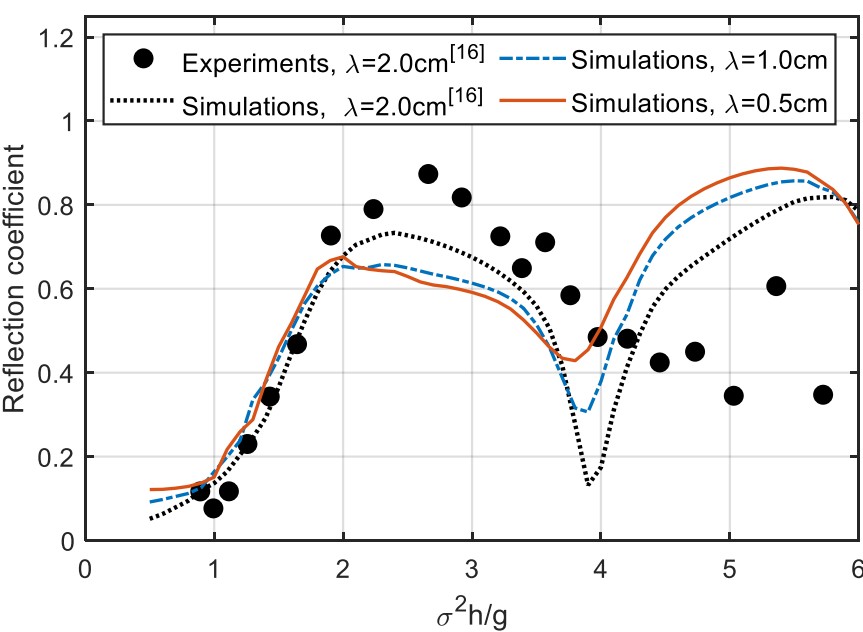

**Figure 6.** Comparison of the reflection coefficient for the platform with various fishnet mesh sizes.

### 4.2. Time-Domain Results

In this section, the time-domain results at the resonant frequency of heave motion ($\sigma^2 h/g = 3.9$) are chosen for discussion. This is because the main difference in the frequency-domain results occurs here.

Figures 7–9 show variations in the surge, heave, and pitch motions of the platform with different fishnet mesh sizes ($\lambda$) during the last 10 waves. All of the results reach steady-state conditions, which means that the simulation results are convergent. In the comparison of surge motion, the results show that both the response amplitude and mean value increase as $\lambda$ decreases. In addition, a short time delay appears as $\lambda$ decreases. In the comparison with the heave motion, the results indicate that the peak value of the vibration decreases greatly with $\lambda$, while the valley value does not. This causes the response amplitude of the heave motion to decrease as $\lambda$ decreases. In the comparison of the pitch motion, the results show that both the peak value and valley value of the vibration decrease as $\lambda$ decreases, but the peak value demonstrates a larger reduction. Thus, the response amplitude of the pitch motion decreases with $\lambda$. In addition, as $\lambda$ decreases, an obvious time delay observed in the pitch motion.

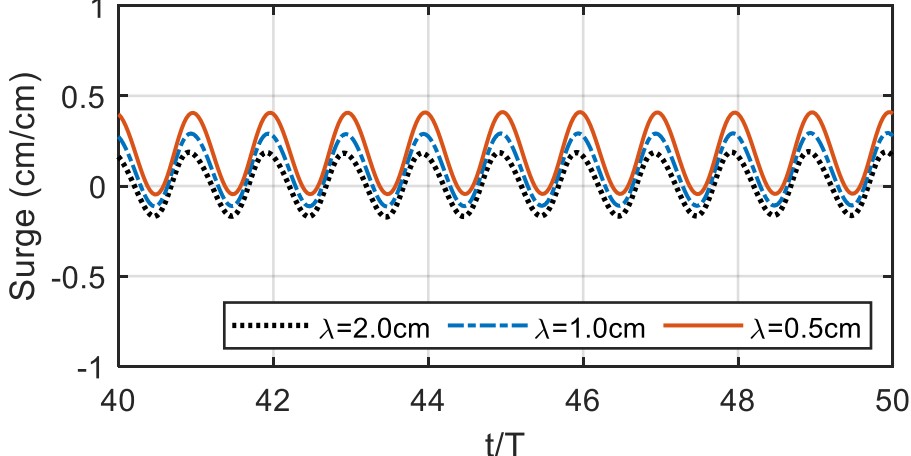

**Figure 7.** Variations in the surge motion for the platform with various fishnet mesh sizes ($H = 4$ cm, $T = 0.91$ s, and $\sigma^2 h/g = 3.9$).

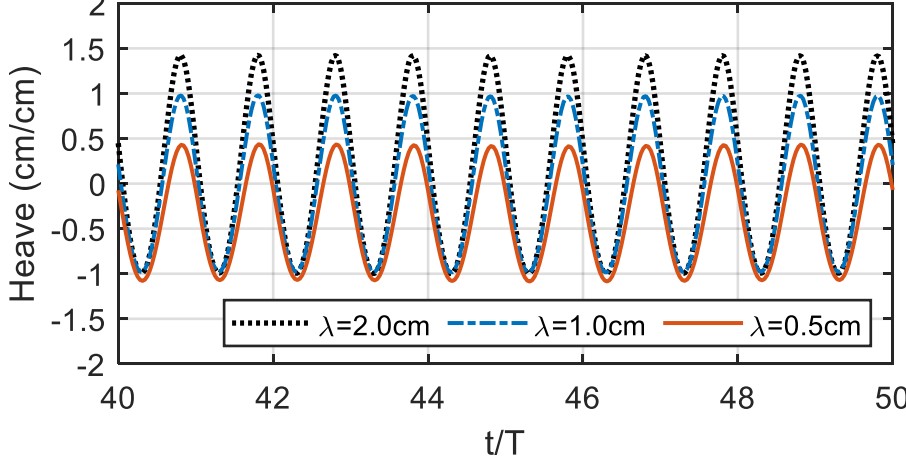

**Figure 8.** Variations in the heave motion for the platform with various fishnet mesh sizes ($H = 4$ cm, $T = 0.91$ s, and $\sigma^2 h/g = 3.9$).

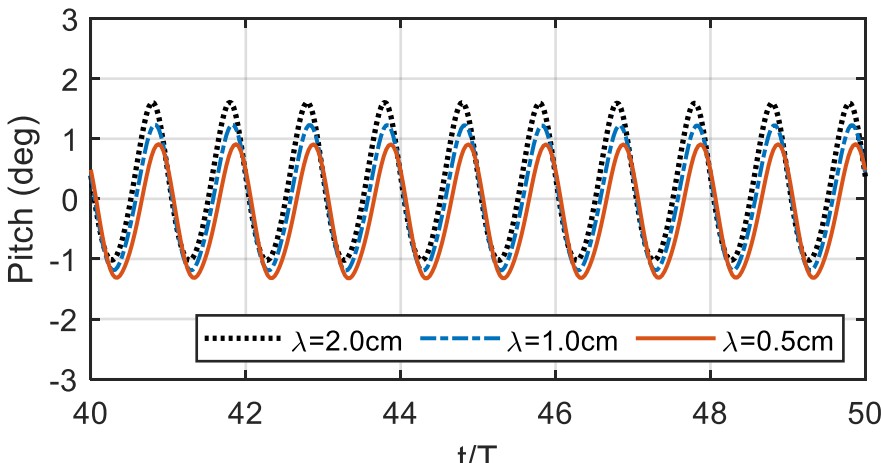

**Figure 9.** Variations in the pitch motion for the platform with various fishnet mesh sizes ($H$ = 4 cm, $T$ = 0.91 s, and $\sigma^2 h/g$ = 3.9.

Figure 10 displays a comparison of sea-side tension for different fishnet mesh sizes ($\lambda$). The results show that as $\lambda$ decreases, the peak value of the tension increases, while the change in the valley value is not obvious. In addition, as $\lambda$ decreases, a slight time delay occurs. Comparing the results to the surge-heave-pitch motions, the phase of peak and valley values of tension are consistent with the phase of heave and pitch motion, but inconsistent with the phase of surge motion. This may be because of the surge motion being associated with restoring mooring force (wave drift forces) rather than wave forces.

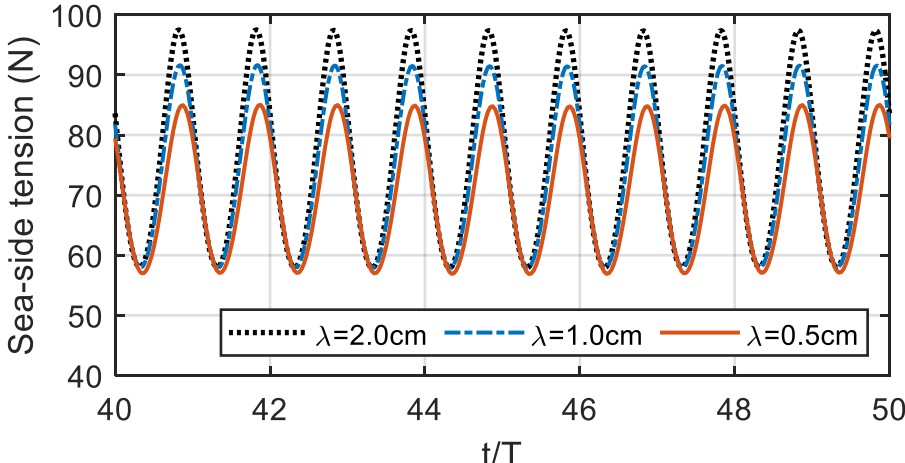

**Figure 10.** Variations in the sea-side tension for the platform with various fishnet mesh sizes ($H$ = 4 cm, $T$ = 0.91 s, and $\sigma^2 h/g$ = 3.9).

Figure 11 illustrates a comparison of wave elevation at gauge P4 (see Figure 1b) for different fishnet mesh sizes ($\lambda$). The results show that as $\lambda$ decreases, the wave amplitude is greatly reduced, which means many incident waves do not pass through the platform as $\lambda$ decreases. Instead, they reflect why the reflection coefficient increases greatly at $\sigma^2 h/g$ = 3.9 in Figure 6.

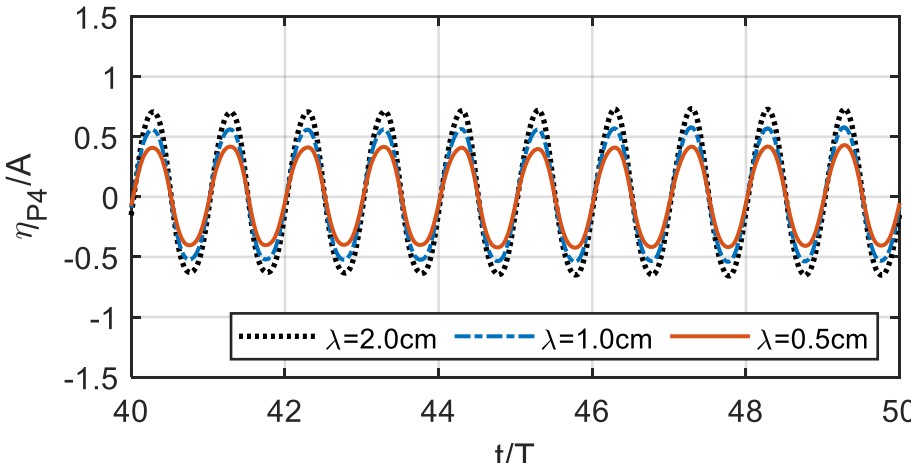

**Figure 11.** Variations in the wave elevation at gauge P4 for the platform with various fishnet mesh sizes ($H$ = 4 cm, $T$ = 0.91 s, and $\sigma^2 h/g$ = 3.9).

### 4.3. Nonlinear Dynamic Properties

In this study, the input wave height is 4 cm, and the wave steepness (wave height/wavelength) is in the range of 0.006 to 0.048. In fact, in this situation, the nonlinearity of waves is not significant. However, in our previous study [16], the second-order RAOs are still easy to observe at the resonant frequencies of the mooring system and platform. Thus, in this section, the second-order RAO of the platform motion and mooring system under different mesh size will be discussed.

Figures 12–14 show the second-order RAO of the surge, heave, and pitch motions of the platform with different fishnet mesh sizes ($\lambda$). In general, the second-order RAO is much smaller than the first-order RAO. However, it can still be observed, especially near the resonant frequencies of the platform and mooring system. Firstly, in the comparison of second-order surge RAO, as $\lambda$ decreases, the RAO increases near the resonance frequency of the mooring system (between $\sigma^2 h/g$ = 1.0 and 2.0), but no significant change near the resonance frequency of the surge motion ($\sigma^2 h/g$ = 4.1) occurs. Next, compared with the second-order heave RAO, as $\lambda$ decreases, the RAO is greatly reduced near the resonance frequency of the heave motion ($\sigma^2 h/g$ = 3.9). However, near the resonance frequency of the mooring system ($\sigma^2 h/g$ = 1), the changes between RAO and $\lambda$ are irregular. As $\lambda$ decreases from 2.0 cm to 1.0 cm, the RAO greatly decreases and the peak value shifts to a higher frequency. In contrast, as $\lambda$ decreases from 1.0 cm to 0.5 cm, the RAO increases, and the peak value shifts to a lower frequency. Finally, in comparison with the second-order pitch RAO, as $\lambda$ decreases, the peaks around the resonant frequency of the pitch motion ($\sigma^2 h/g$ = 4.1) hardly differ, but the frequency band is wider. Around the resonance frequency of the mooring system (between $\sigma^2 h/g$ = 1.0 and 2.0), it seems that the smaller the mesh size, the lower the RAO.

Figure 15 displays the second-order sea-side tension RAO of the platform with different fishnet mesh sizes ($\lambda$). In fact, the second-order tension RAO is very small by comparison with the first-order tension RAO. The results show that, near the frequency of $\sigma^2 h/g$ = 4.0, the RAO decreases with $\lambda$. Apart from that, the relationship between RAO and $\lambda$ is irregular.

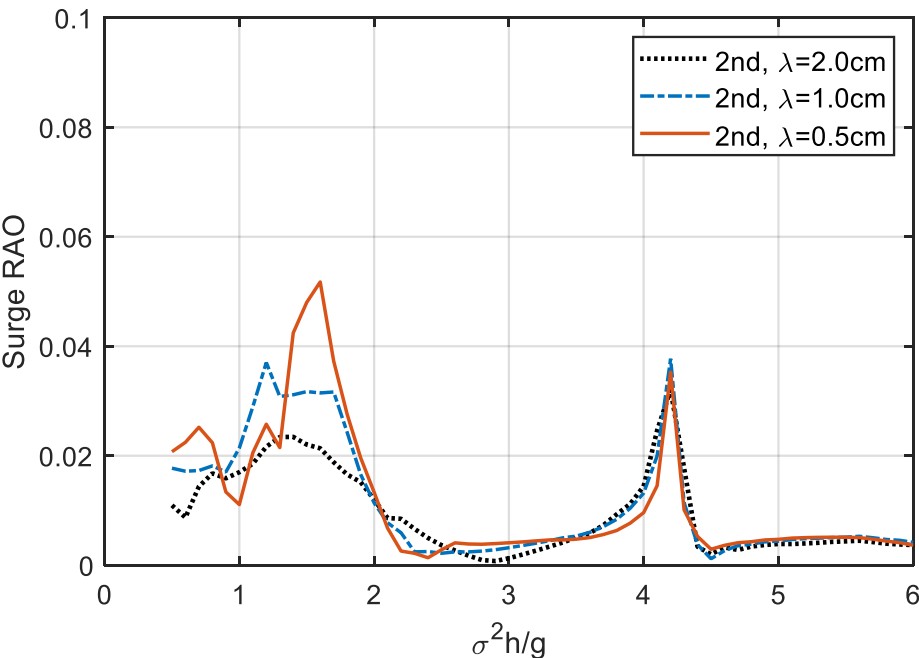

**Figure 12.** Comparison of the second-order surge RAO of the floating platform with different fishnet mesh sizes.

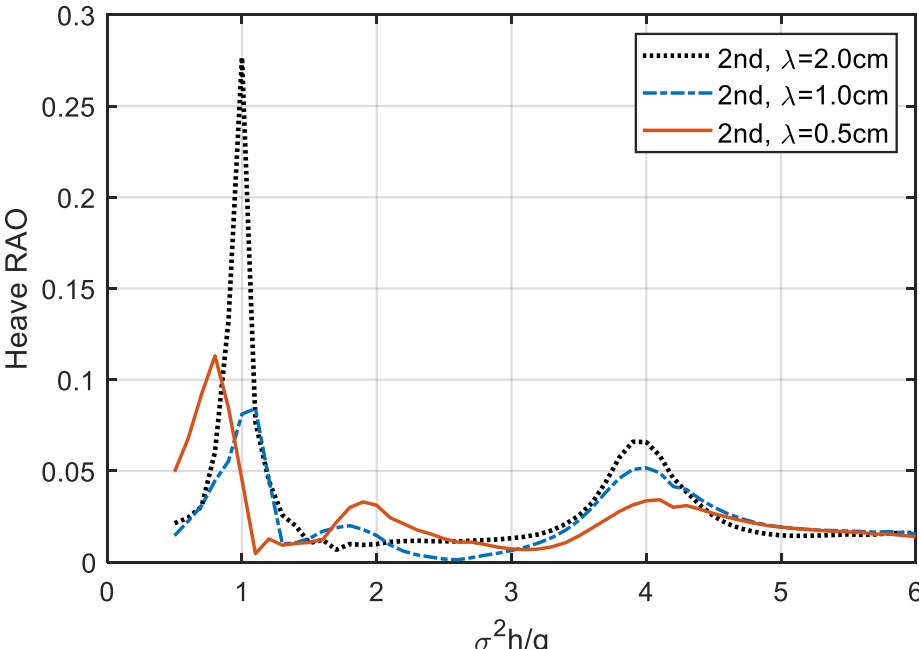

**Figure 13.** Comparison of the second-order heave RAO of the floating platform with different fishnet mesh sizes.

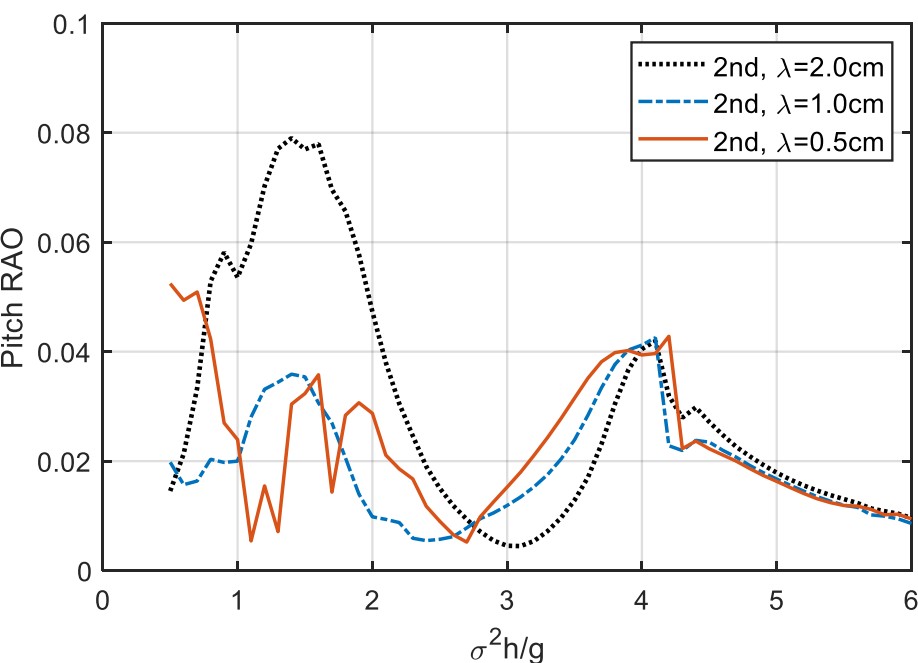

**Figure 14.** Comparison of the second-order pitch RAO of the floating platform with different fishnet mesh sizes.

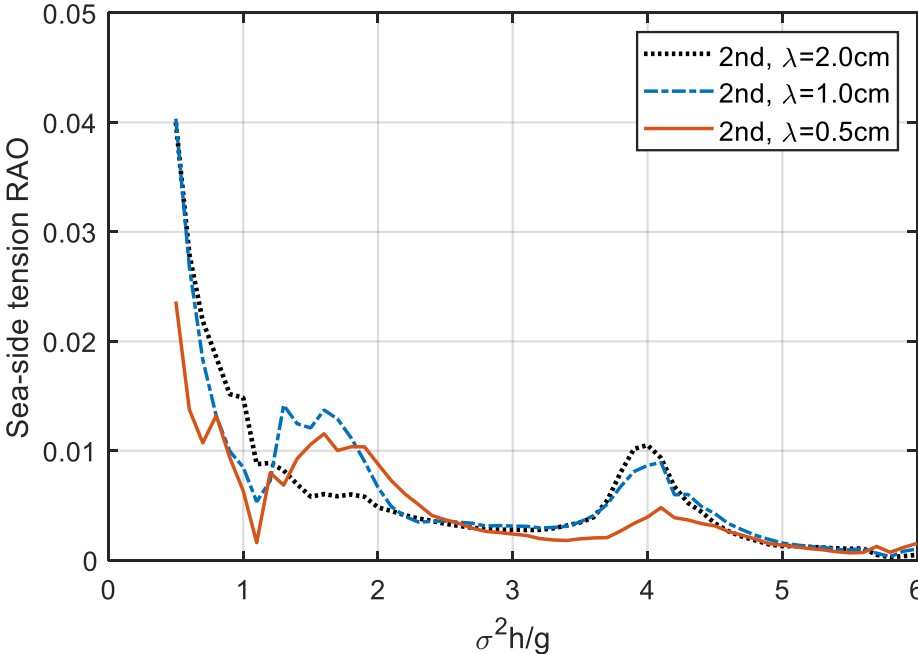

**Figure 15.** Comparison of the second-order sea-side tension RAO of the floating platform with different fishnet mesh sizes.

## 5. Conclusions

In this study, a fully nonlinear numerical wave tank was developed to investigate the dynamic response of a floating platform with three different fishnet mesh sizes. The mooring system was considered to be linearly elongated and symmetrically installed. This model is solved by the BEM, while the free surface nodes were tracked by the MEL approach with a cubic spline scheme and the RK4 method. Damping zones were arranged at both ends of the tank to absorb reflected wave energy

and dissipate the transmitted wave energy. The instantaneous floating body motion was calculated by means of an acceleration potential method and a modal decomposition method.

The frequency-domain results show that as the fishnet mesh size decreases, all of the surge, heave, pitch, and sea-side tension RAO reduce, apart from the pitch and tension RAOs in the lower frequency region. In addition, the most significant reduction is observed at the resonant frequency of heave motion. At this frequency, the time-domain results indicate that major reductions in platform motions, tension forces, and transmitted wave heights will appear when decreasing the fishnet mesh size and may even occur with a slight time delay. Moreover, the second-order RAOs for the platform motion and mooring tension are observed in the simulation. However, these are very small compared with the first-order RAO. Essentially, most of them are decreased with the fishnet mesh size.

In the future, attaching fishnets to any kind of floating platform is not necessarily for aquaculture purposes only. It can also be a damper or breakwater because it can not only stabilize the movement of the structure, but also reduce the impact of waves.

**Author Contributions:** Conceptualization, H.-J.T. and R.-Y.Y.; methodology, H.-J.T. and C.-C.H.; software, H.-J.T.; validation, H.-J.T., C.-C.H. and R.-Y.Y.; formal analysis, H.-J.T.; investigation, H.-J.T.; resources, H.-J.T. and R.-Y.Y.; data curation, H.-J.T.; writing—original draft preparation, H.-J.T.; writing—review and editing, H.-J.T., C.-C.H. and R.-Y.Y.; visualization, H.-J.T.; supervision, R.-Y.Y. and C.-C.H.; project administration, H.-J.T. and R.-Y.Y.; funding acquisition, H.-J.T. and R.-Y.Y. All authors have read and agreed to the published version of the manuscript.

**Funding:** This research was funded by the Ministry of Science and Technology of Taiwan, grant number MOST 109-3116-F-006-013-CC1 and MOST 109-2222-E-006-003-MY2. The APC was funded by the Ministry of Science and Technology of Taiwan (MOST 109-3116-F-006-013-CC1).

**Acknowledgments:** The authors would like to thank the reviewers for their constructive comments and suggestions and the great support from the Ministry of Science and Technology of Taiwan.

**Conflicts of Interest:** The authors declare no conflict of interest.

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
