# Peer review of "Numerical Study of the Influence of Fishnet Mesh Size on a Floating Platform"

_jmse, doi:10.3390/jmse8050343_

Round 1

Reviewer 1 Report

I find the paper an excellent work and very interesting and actual.

Just a curiosity/ suggestion.

After the validation of the results in section4.1 regarding the frequency domain analysis, why the authors don't use the experimental data also in the time domain analysis and add them in the following section on time domain analysis and in next plots.

I think it could also improve the work if not possible to use them(why?)  I suggest to discuss why the data are not used in the time domain analysis.

Reviewer 2 Report

This paper studied the validation of previous work of [16] (based on numerical BEM) by physical model test and also explained what is new did in the present manuscript. Although, I’ve not checked all the equations but they seem to be OK. The present work has interest in the area of marine aquaculture to develop an aquaculture platform for fish cage. However, the reviewer writes some comments to the author to act upon:

1) In Eq. (1), the author needs to mention two-dimensional Laplace equation and define the Laplacian operator, 2) In Eq. (17), could you please clarify why a constant is not associated with diffraction mode (φ4) ?, 3)  Throughout the manuscript, the author is missing “period” and “commas” at the end of the equations, 4)The numerical results are well explained and of good quality, 5) Please mention the future scope of the present study at the end of the conclusion, 6) The recent development of net-type structures application to fish cages was reviewed in the following paper which may be interest to the present study:

  • C. Guo, S.C. Mohapatra & C. Guedes Soares.  2020. Review of developments in porous membranes and net-type structures for breakwaters and fish cages. Ocean Engineering, 200: 107027.

Round 2

Reviewer 2 Report

The authors have revised the manuscript based on my comments. Therefore, now the revised manuscript can be considered for publication.